# Quantitative Evaluation of Interleukin-4 by Immunowall Devices Made of Gelatin Methacryloyl Hydrogel

**DOI:** 10.3390/molecules28124635

**Published:** 2023-06-08

**Authors:** Yuto Banno, Takuma Nomiyama, Shoma Okuno, Sachiko Ide, Noritada Kaji

**Affiliations:** Department of Applied Chemistry, Graduate School of Engineering, Kyushu University, 744 Motooka, Nishi-ku, Fukuoka 819-0395, Japan

**Keywords:** immunoassay, lab on a chip, interleukin, gelatin methacryloyl hydrogel

## Abstract

Immunoassays, which use antigen–antibody reactions, are the primary techniques used to selectively quantify specific disease markers in blood. Conventional immunoassays, such as the microplate-based enzyme-linked immunosorbent assay (ELISA) and paper-based immunochromatography, are widely used, but they have advantages and disadvantages in terms of sensitivity and operating time. Therefore, in recent years, microfluidic-chip-based immunoassay devices with high sensitivity, rapidity and simplicity, which are compatible with whole blood assays and multiplex assays, have been actively investigated. In this study, we developed a microfluidic device using gelatin methacryloyl (GelMA) hydrogel to form a wall-like structure in a microfluidic channel and perform immunoassays inside the wall-like structure, which can be used for rapid and highly sensitive multiplex assays with extremely small sample amounts of ~1 μL. The characteristics of GelMA hydrogel, such as swelling rate, optical absorption and fluorescence spectra, and morphology, were carefully studied to adapt the iImmunowall device and immunoassay. Using this device, a quantitative analysis of interleukin-4 (IL-4), a biomarker of chronic inflammatory diseases, was performed and a limit of detection (LOD) of 0.98 ng/mL was achieved with only 1 μL sample and 25 min incubation time. The superior optical transparency over a wide range of wavelengths and lack of autofluorescence will help to expand the application field of the iImmunowall device, such as to a simultaneous multiple assay in a single microfluidic channel, and provide a fast and cost-effective immunoassay method.

## 1. Introduction

Immunoassays, which use antibodies with a high reactivity to specific antigens, are widely used to selectively quantify specific disease markers in blood. In particular, microplate-based ELISA, such as sandwich ELISA, has been conventionally used as a highly sensitive and quantitative immunoassay for biomarkers related to chronic inflammatory diseases [1]. Although this method has a relatively high quantitative performance, several issues have been raised, including the long preparation and measurement time of several hours, the need for skilled operators, the need for large and expensive equipment, the use of large amounts of reagents for measurement, and the lack of high sensitivity in some cases [2]. In addition, paper-based immunoassays such as immunochromatography are widely used as a simpler measurement method [3]. This method uses capillary action to move the sample across a nitrocellulose membrane, and antigen–antibody reactions are performed by reagents prepared on the cellulose membrane. Although this method is simple, fast, and inexpensive to use, it has the problem of sensitivity. Therefore, various immunoassay devices using microfluidic devices have been developed to minimize the disadvantages and achieve high sensitivity, rapidity and simplicity, perform the assay at a low cost, perform the assay in whole blood, and simultaneously diagnose multiple biomarkers from the same sample [4,5].

Immunowall devices are devices in which a wall-like structure is created in the center of a straight microfluidic channel, parallel to the channel, using a light-curable material [6]. By introducing GelMA solution, a photo-curable material, into the microfluidic channel and irradiating it with ultraviolet light through a photomask, a wall can be created in the channel because only the area exposed to the light is cured [7]. The high surface-to-volume ratio allows for sandwich immunoassays to be performed on the wall surface. The high surface-to-volume ratio allows for shorter incubation times for the assay. In a previous study, a quantitative analysis of EGFR mutations in tumor tissue achieved a detection limit of 0.01 mg/mL in less than 30 min of operation [8]. Operation requires only pipetting and does not require difficult techniques, as the liquid is transferred by capillary forces, and the chip itself is disposable and could avoid cross-contamination during the immunoassay. In addition, the Immunowall device met many of the requirements of recent microfluidic device research for immunoassays, such as using only about 1 µL of reagent in a single assay. When quantifying multiple biomarkers from the same sample, it is necessary to use different types of fluorescent labels at different wavelengths to avoid overlapping fluorescence signal wavelengths. However, azide-unit pendant water-soluble photopolymer (AWP) (TOYO GOSEI), a photocurable material for walls used in the previous study, has autofluorescence in the visible light region, which interferes with the signals and is not suitable for multiple assays [9]. Therefore, it is important to develop an immunowall device using a photocurable material that does not have autofluorescence in the visible-light region.

In this study, we focused on GelMA hydrogel, a light-curing material that does not exhibit autofluorescence in the visible-light region. GelMA has a structure in which an amino group in the protein chain of gelatin is replaced by a methacrylamide group by reaction with methacrylic anhydride. 2-Hydroxy-4′-(2-hydroxyethoxy)-2-methylpropiophenone (Irgacure 2959) or other photoinitiators cure GelMA to a hydrogel and provide a high swelling performance due to its hydrogel nature, allowing for it to be used as an immunoassay reaction platform. The high swelling performance of GelMA hydrogel also allows for the sufficient penetration of reagents into the wall during immunoassays. In addition, because it is widely used as a scaffold for cell culture, it has high biocompatibility and is considered to have high affinity for whole blood immunoassays [10]. Therefore, GelMA hydrogel is considered highly suitable as a wall material for immunowall devices. IL-4, which is a cytokine and an important biomarker of inflammatory diseases such as infectious diseases [11] and chronic inflammation resulting from autoimmune diseases [12,13], was set as the target of this study. It has a wide range of blood levels even under healthy conditions, so frequent monitoring is required for the early diagnosis of chronic inflammatory diseases. Since these diseases are difficult to treat after their onset, it is necessary to assess the immune status of the body in advance and receive early medical treatment. In the future, not only IL-4, but also IL-6, IL-12, and TNF-α, which are also required for blood level monitoring, should be analyzed by multiple immunoassay systems using our Immunowall device. GelMA, which has superior optical transparency over a wide range of wavelengths, may be suitable to perform the multiple immunoassays in the Immunowall simultaneously and is expected to save assay time and provide a cost-effective method.

## 2. Results and Discussion

### 2.1. Characterization of GelMA Hydrogel

The absorbance of the photopolymerized hydrogels at 380~780 nm was measured using a microplate reader, and the results are shown in Figure 1a. Overall, the absorbance was lower than that of AWP. Fluorescence measurements were also performed at excitation wavelengths of 360 nm, 470 nm, 545 nm and 620 nm, as shown in Figure 1b–e. The fluorescence at around 450 nm and 550 nm was dramatically suppressed when excited at 360 nm and 420 nm, respectively, and the low autofluorescence from the GelMA hydrogel proved to be the wide application of different colored fluorescent dyes for simultaneous multiple assays. The results of the swelling measurement are shown in Figure 1f. The swelling was almost complete within one hour after immersion in the buffer solution, and the weight swelling ratio after 24 h was 8.73. Therefore, this material is considered to have the ability to sufficiently incorporate and exchange antigen and antibody solutions into the wall structure for quantitative immunoassay.

### 2.2. Fabrication and Characterization of GelMA Immunowalls

The walls were fabricated in the microfluidic channel by UV exposure for 30 s, and the average size of the five fabricated walls was 40.1 µm in length and 3.98 µm in width, indicating that the walls were cured almost according to the photomask design. After the washing process, the length and width were slightly increased to 40.2 µm and 4.52 µm, respectively, which may be due to the swelling of the GelMA hydrogel wall caused by the washing process. Microscopic observations after wall fabrication and immunoassay showed that the whole wall was deformed and had a helical structure. This phenomenon may be caused by the detachment of GelMA hydrogel from the microfluidic channel induced by swelling, and prevents quantitative evaluation of the fluorescence-based immunoassay. To elucidate the cause of this phenomenon, two washing processes, with or without drying of the wall, were performed after UV irradiation, and the wall structure was observed by microscopy. As shown in Figure 2, this deformation was observed only when the hydrogel underwent a re-swelling process after drying, and no deformation was observed when the wall remained wet. These results also suggested that the surfactant in the washing solution or the salt in PBS did not affect the wall structure. Therefore, the immunowall was kept wet in solution until immunoassay. We have not yet studied the long-term performance retention of the immunowalls over one week, but the stability of the immunoassay reactivity in the immunopillars has been confirmed over 3 months in refrigerator in our previous work [5]. Therefore, even in the immunowalls, a similar period of time can be used for the immunoassay if no deformation was observed associated with evaporation.

In this immunoassay system, streptavidin was used as the starting point for subsequent antibody immobilization and antibody–antigen reactions within the polymer network. To evaluate the capture and retention ability of GelMA hydrogel, a 1.0 μm diameter fluorescent bead was premixed with GelMA hydrogel and cured by UV in the microfluidic channel. As shown in Appendix A, the fluorescent bead remained inside the wall even after 10 repeated washes, and no leakage was observed. Therefore, the average pore size might be less than 1.0 μm and can hold the immunocomplex (c.a. ~100 nm) during the immunoassay process even under extensive washing.

### 2.3. Immunoassay of IL-4

The immunoassay of IL-4 was performed, the calibration curve was plotted as shown in Figure 3, and the calculated LOD was 0.98 ng/mL. Since the cut-off value of IL-4 in blood is about 6 pg/mL in clinical use, the lower detection limit (LOD) needs to be improved by optimizing the reaction condition. In this immunoassay, even at an IL-4 concentration of 0 ng/mL, slight fluorescence in the wall was observed by fluorescence microscopy, as shown in Figure 3a. Since the wall without streptavidin did not show any fluorescence, it is likely that the streptavidin is responsible for this phenomenon. Since this phenomenon was observed in another assay using IL-6, it is unlikely that the reagent itself or the compatibility of the reagent with the wall were the cause of this, but the streptavidin itself could be the cause. The same phenomenon was observed when the UV exposure device for the wall preparation was changed and the assay was performed, indicating that the curing method used for the hydrogel was not the cause of this phenomenon. The low fluorescence conditions with no antigen (IL-4) are critical to achieve a highly sensitive immunoassay. Therefore, to elucidate the cause of this phenomenon, we performed the assay with the combination of reagents shown in Appendix A. The results are shown in Appendix A. Fluorescence was observed in the case of a, c, and d. This result may indicate that the secondary or tertiary antibody binds directly to the streptavidin pre-mixed in GelMA hydrogel without the primary or secondary antibody. This non-specific binding of antigen and antibody is often observed in immunoassays. Therefore, a further investigation was carried out by changing the antibody lot, and dissociation constant measurements using Biacore are required to improve the LOD in this immunoassay. The detection of biomarkers of chronic inflammatory diseases should be performed from serum or whole blood samples for future practical diagnosis. Therefore, the interference problems caused by abundant proteins such as albumin and blood cells should be carefully investigated to improve the LOD. A prefilter that is directly connected to the inner reservoir could provide the solution and was developed to remove blood cell components before immunoassay. In this study, the entire assay was performed at room temperature, but for future practical application, quality control of the device at different assay temperatures, humidities, and preservation conditions should be addressed.

## 3. Materials and Methods

### 3.1. Preparation and Characterization of GelMA Hydrogel

At a final concentration of 0.05 g/mL of gelatin methacrylate (300 bloom, degree of substitution 60%) (Tokyo Chemical Industry Co., Ltd. (TCI), Tokyo, Japan), 0.01 g/mL of 2-Hydroxy-4′-(2-hydroxyethoxy)-2-methylpropiophenone (Irgacure-2959, Sigma-Aldrich Japan K.K., Tokyo, Japan), and 5 mg/mL of Streptavidin (recombinant, expressed in *E. coli*, Sigma-Aldrich Japan K.K., Tokyo, Japan) were mixed in phosphate-buffered saline (1 × PBS) at 50 °C on a hot plate while stirring occasionally with a vortex mixer. To characterize GelMA hydrogels, 200 µL of the solution was poured into a transparent 96-well microtiter plate and the solution was cured by UV irradiation for 30 s in a UV exposer (LA-410UV-5; Hayashi-Repic Co., Ltd., Tokyo, Japan) (Figure 4). For absorbance measurements, the hydrogels were prepared in transparent 96-well microplate and the absorbance in the wavelength range of from 380 to 780 nm was measured using a microplate reader (Infinite 200 PRO M Plex, Tecan Japan Co., Ltd., Kanagawa, Japan). The fluorescence measurements were performed in black 96-well microtiter plates with a clear flat bottom at excitation wavelengths of 360 nm, 470 nm, 545 nm and 620 nm using the microplate reader. The same measurements were also performed for AWP and 1 × PBS as background. Swelling measurements were performed as follows: hydrogels prepared in 96-well microplates were removed onto a disposable petri-dish using a spatula and freeze-dried at −80 °C for 24 h using a lyophilizer (FDU-2200, Tokyo Rikakikai Co., Ltd., Tokyo, Japan). The hydrogels were weighed after lyophilization and immersed in 1 × PBS at 25 °C to swell.1, 2, 3, 4, 5, 6, and 24 h later, the hydrogels were weighed by gently wiping off the surface water with a Kimwipe. The weight swelling ratio was calculated by dividing the measured weight by the lyophilized weight.

### 3.2. Fabrication of Microfluidic Devices

As shown in Figure 5, the substrate for the microfluidic device in this study was made of cycloolefin copolymer (COC, Sumitomo Bakelite Co., Ltd., Tokyo, Japan) of 30 mm long, 70 mm wide, and 1.4 mm thick, consisting of inlets, outlets, and microfluidic channels. The inlet and outlet diameters are 1.0 mm, and each device has 40 microfluidic channels of 1 mm wide, 0.05 mm deep, and 8.5 mm long. The chromium photomask has 24 rectangular slits of 4.0 mm long and 40 μm wide are designed to overlap in the center of the microfluidic channels when the photomask overlaps on the device. A total of 1 µL of GelMA solution was poured into the inlet and introduced into the channel by aspiration from the outlet using a syringe pump (YSP-202, YMC, Kyoto, Japan). Normally, solution introduction into the channel in this study is carried out by capillary force only [5], but because of the relatively high viscosity of the GelMA solution, gentle aspiration accelerated this process and saved time. GleMA hydrogel was cured into the wall shape in the microfluidic channel by UV exposure through the photomasuk. Immediately thereafter, uncured solution was washed away by aspiration and a washing solution was introduced from the inlet to fill the channels and aspirated from the outlet using a portable aspirator (VACUSIP, INTEGRA Biosciences Corp., Hudson, NH, USA). This washing process was performed 10 times for each channel. After the washing, the channels were filled with the washing solution, the inlet and outlet were sealed with a tape to prevent drying, and the microfluidic devices were wrapped with aluminum foil and stored in a refrigerator at 4 °C.

The washing solution was prepared by mixing 1% of bovine serum albumin (BSA, Thermo Fisher Scientific Inc., Tokyo, Japan) (1% BSA-PBS) and 1% of Tween^®^ 20 (Sigma-Aldrich Japan K.K., Tokyo, Japan) in 1 × PBS at 1-to-1 ratio (final concentration: 0.5% BSA and 0.5% Tween^®^ 20 in 1 × PBS). This washing solution was also used for washing process during immunoassays and the 1% BSA-PBS was also used for the dilution of reagents used in immunoassays. Phase contrast images of the fabricated walls were taken with a microscope (ECLIPSE Ts2, Nikon Solutions, Co., Ltd., Tokyo, Japan).

### 3.3. Immunoassay

A total of 1 μL of biotin anti-human IL-4 antibody (50 µg/mL) (BioLegend, Inc., San Diego, CA, USA) was introduced into the microfluidic channel and incubated for 60 min to bind to streptavidin, which was pre-mixed in the GelMA hydrogel wall. After washing with the washing buffer, 1 µL of recombinant human IL-4 protein (Proteintech Group, Inc., Tokyo, Japan) was introduced and incubated for 15 min to react with the primary antibody. After washing with washing buffer, 1 µL of secondary antibody, anti-IL-4 antibody (from 0.156 μg/mL to 10 ng/mL, rabbit polyclonal) (Abcam plc., Cambridge, UK) was introduced and incubated for 5 min to react with the antigen. After washing with washing buffer, 1 µL of tertiary antibody, Alexa Fluor^®^ 488 labelled Goat anti-rabbit pre-adsorbed Ig-G (50 µg/mL) (Abcam plc.) was introduced and incubated for 5 min to react with the secondary antibody. The tertiary antibody binds specifically to the secondary antibody by cross-reactivity and is labeled with a fluorescent dye. Finally, the microfluidic channels were washed with the washing buffer. This forms a streptavidin-biotin-labeled primary antibody–antigen-secondary antibody–fluorescent-labeled tertiary antibody complex. During the incubation and after the final wash, the inlet and outlet were sealed with tape to prevent from drying. As a negative control, the wall without streptavidin was prepared and operated in the same manner with no antigen. All reagents used here were gently agitated in a vortex mixer and then spun down in a tabletop centrifuge immediately before use.

The entire washing process was performed by the following operations: the reagents that finished incubation were aspirated with the portable aspirator, and 1 µL of the washing solution was introduced to the microfluidic channel, followed by 1 min incubation. Then, the washing solution was aspirated with a portable aspirator, and 1 µL of washing solution was introduced again. After the last washing process, the wall was observed using a fluorescence microscope (IX83, Evident Corp., Tokyo, Japan). The appropriate laser and filter sets were selected for the fluorescent dye on the tertiary antibody, and a camera (ImagEM X2 C9100-23B, Hamamatsu Photonics K.K., Hamamatsu, Japan) was used to take images of the wall. An objective lens of 10× was set and the exposure time was selected as appropriate. Images of the wall taken with the fluorescence microscope were analyzed using PC software (cellSens imaging software, ver. 3.2.23706.6 (accessed on 13 October 2022), Evident Corp., Tokyo, Japan) bundled with the fluorescence microscope. The bright areas on both sides of the wall were enclosed in squares and their average luminance was measured, as shown in Figure 5. In order to eliminate artificial prejudice during the analytical process, the measured rectangle area was set in almost the same position upstream, midstream, and downstream. (The wall could not fabricate exactly the same position in the microfluidic channel. There is always a slight inclination against the channel.) The fluorescence intensities from both sides of the wall upstream, midstream, and downstream were measured and the average fluorescent intensity was plotted as a function of IL-4 concentration. The limit of detection (LOD) was calculated from the assay results of 0 ng/mL of IL-4 using the 3σ method.

## 4. Conclusions

In this study, we fabricated an immunowall device using GelMA hydrogel to create a wall-like structure in a microfluidic channel and performed sandwich immunoassays on and inside the structure for the rapid quantitative analysis of IL-4, a biomarker of inflammatory diseases. The GelMA hydrogel wall has no autofluorescence, making this microfluidic device more suitable for multiplex assays using different fluorescently labeled antibodies to conventional UV-curable immunowalls. Although further optimization of the device, reagent conditions, incubation time and washing method is required to improve LOD and reproducibility, the device is expected to be applied in the rapid and cost-effective quantitative analysis of biomarkers, not only for inflammatory diseases but also for other diseases. The new applications of GelMA hydrogel might expand the application field from biochemical research to practical diagnosis and explore its possibile use as a biocompatible material.

## Figures and Tables

**Figure 1 molecules-28-04635-f001:**
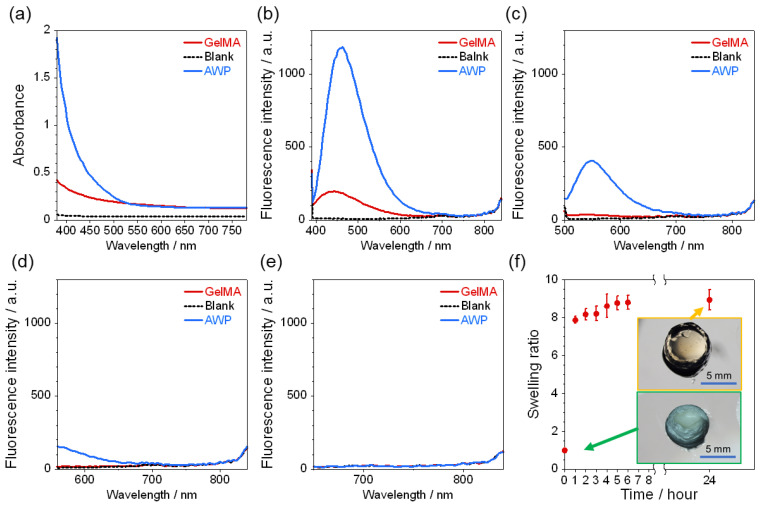
(**a**) Absorption and fluorescence spectrum of GelMA and AWP hydrogel at (**b**) 360 nm, (**c**) 470 nm, (**d**) 545 nm, and (**e**) 620 nm excitation. (**f**) Swelling ratio of GelMA hydrogel as a function of time. The inserted images show photos at 0 and 24 hrs indicated by the arrows, respectively.

**Figure 2 molecules-28-04635-f002:**
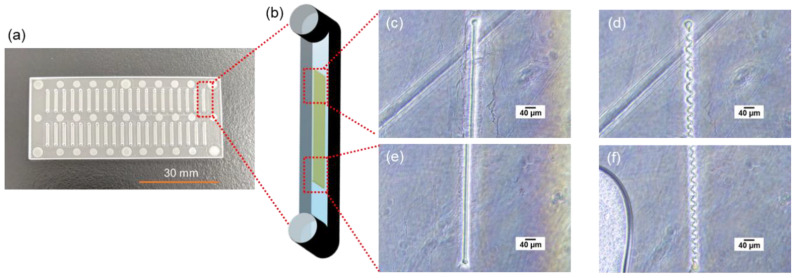
(**a**) An overview photo of the microfluidic chip with 70 mm × 30 mm square and 1.4 mm thick COC plate. (**b**) A schematic illustration of a microfluidic channel 8.5 mm long, 1 mm wide, and 50 μm deep. A GelMA hydrogel wall structure 4.0 mm long, 40 μm wide, and 50 μm tall was fabricated inside the microfluidic channel. Bright field microscope images of (**c**,**e**) the upper and lower regions of the wall after drying and (**d**,**f**) after re-swelling by introducing the solution to the dried state shown in (**c**,**e**), respectively.

**Figure 3 molecules-28-04635-f003:**
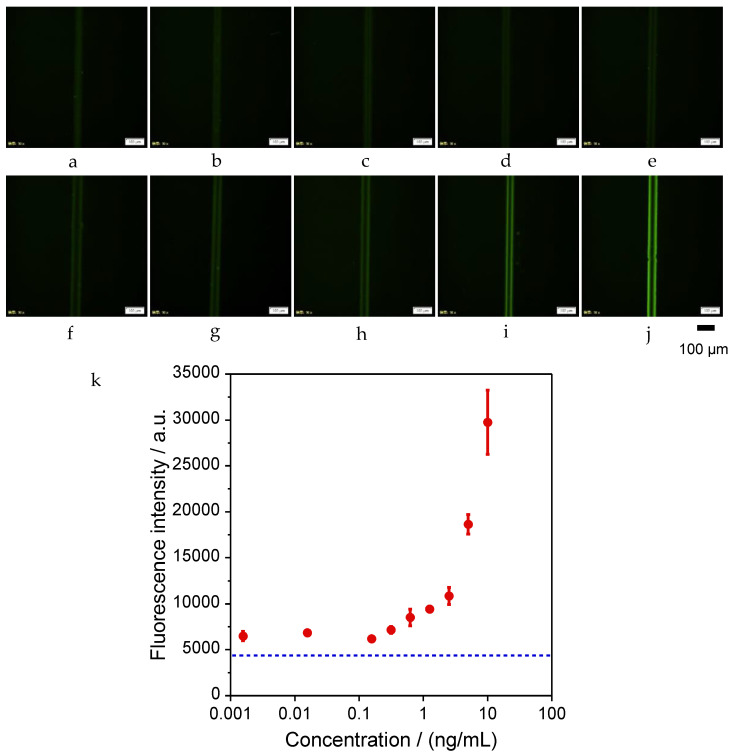
Fluorescence microscope images of immunoassay at IL-4 concentration of (**a**) 0, (**b**) 1.56 × 10^−3^, (**c**) 15.6 × 10^−3^, (**d**) 0.156, (**e**) 0.313, (**f**) 0.625, (**g**) 1.25, (**h**) 2.50, (**i**) 5.00, (**j**) 10.0 ng/mL. (**k**) Calibration curve of IL-4 immunoassay. The dotted line indicates the fluorescence intensity of negative control measured by a fluorescence image of only the wall without any reagents.

**Figure 4 molecules-28-04635-f004:**
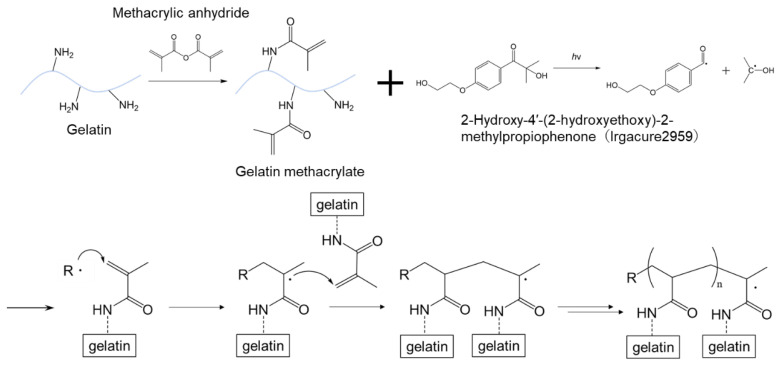
Reaction scheme of Gelatin methacrylate and Irgacure2959 for GelMA hydrogel formation.

**Figure 5 molecules-28-04635-f005:**
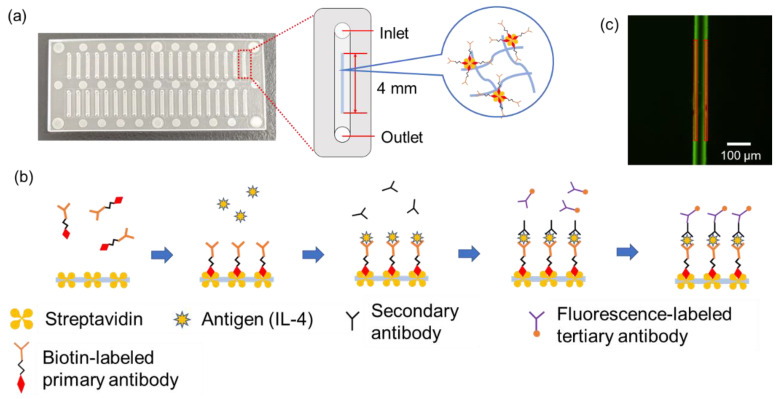
(**a**) Experimental scheme of GelMA hydrogel formation and (**b**) immunoassay inside microfluidic channel. (**c**) The final product in the wall, fluorescence-labeled immunocomplex, was recorded by fluorescence microscope as an image and the fluorescence intensity of both wall edges were measured.

## Data Availability

The datasets used and/or analyzed during the current study available from the corresponding author on reasonable request.

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
