# Peer review of "Quantitative Evaluation of Interleukin-4 by Immunowall Devices Made of Gelatin Methacryloyl Hydrogel"

_molecules, 2023, doi:10.3390/molecules28124635_

Round 1

Reviewer 1 Report

In this manuscript, the authors reported a microfluidic device using gelatin methacryloyl hydrogel to form a wall-like structure in a microfluidic channel and perform immunoassays inside the wall-like structure. the calibration curve was plotted and the calculated LOD was 0.98 ng/mL with only 1 μL sample and 25 minutes incubation time, demonstrating its wide usability in practical applications. However, I have several concerns before this manuscript can be accepted. Therefore, in its current form, revisions are needed.

1. Will the assay results be influenced or polluted by internal and foreign matter? Since the sensitivity is an important technical feature of the chip, how is the specificity of this detection method?

2. Will the humidity and temperature affect the results? How?

3. How about the repeatability and reusability of this chip?

Author Response

Comment           

1) Will the assay results be influenced or polluted by internal and foreign matter? Since the sensitivity is an important technical feature of the chip, how is the specificity of this detection method?

Response

As the reviewer pointed out, coexisting matters affect the assay results. Our final goal is to detect biomarkers related to chronic inflammatory diseases from whole blood, so abundant proteins such as albumin and blood cells might interfere immunoreaction or cause microchannel clogging. Although we did not show immunoassay results using IL-4 spiked human serum sample, the current LOD is 3.16 ng/mL, which is almost three times worse than the sample (PBS) used in this study. Therefore, the description about several trials to guarantee the LOD are included in the Results and Discussion.

Comment

2) Will the humidity and temperature affect the results? How?

Response

In this study, all the experiment performed under the room temperature and no control of humidity. Regarding the humidity, GelMA hydrogel kept wet by PBS before use. So it might be no problem for the immunoreaction. We have not yet studied long-term performance retention of the immunowalls over one week, but the stability of the immunoassay reactivity in the immunopillars has been confirmed over 3 months in our previous work.(Reference 5) Therefore, even in the immunowalls, a similar period of time can be used for the immunoassay if no deformation was observed during associated with evaporation. The additional description was included in the Results and Discussion.

Comment

3) How about the repeatability and reusability of this chip?

Response

If my understanding of your comment is correct, the COC microfluidic chip itself is disposable and do not use again. The immunoassay repeatability using different chip was quite good and the calibration curve and the LOD always the same value even in the different chip.

Reviewer 2 Report

The manscript describes the development of microfluidic device for the immunoassay of IL-4 made from gelatin methacryloyl hydrogel. I appreciate your efforts for data prerentation. The manuscript is well-prepared and clear. But it cannot be recommended for publication due to lack of scientific significance and low novelty. The described experiment is a kind of preliminary studies but not a sound study suitable for full research paper preparation. Gelatin methacryloyl hydrogel is not a new material. The detection of IL-4 with the described device is insufficient. Regardless the poor detection limit, the biosensor always should be characterized more extensively.

Author Response

Comment           

The manscript describes the development of microfluidic device for the immunoassay of IL-4 made from gelatin methacryloyl hydrogel. I appreciate your efforts for data prerentation. The manuscript is well-prepared and clear. But it cannot be recommended for publication due to lack of scientific significance and low novelty. The described experiment is a kind of preliminary studies but not a sound study suitable for full research paper preparation. Gelatin methacryloyl hydrogel is not a new material. The detection of IL-4 with the described device is insufficient. Regardless the poor detection limit, the biosensor always should be characterized more extensively.

Response

Thank you for your valuable comment. We agree that our system showed insufficient LOD for IL-4 at current status. However, we thought that this study showed the first steps with different hydrogel for immunoassay and explore the new application field. We believe that the study might worth to report in this journal.

Round 2

Reviewer 1 Report

The authors have not done the best job of answering these questions. It is better to offer the experimental evidence, but they did not provide any experimental data. 

Reviewer 2 Report

Dear authors,

If you wish to focus on microfluidic system formation, you should strongly reorganize your manuscript and to explain its novelty.

If you wish to focus on IL-4 detection, than an appropriate sensor characterization research should be carried out. 

I cannot recomment the manuscript for publication in its current version.